# Genome-Wide Association Study Identified Novel Candidate Loci/Genes Affecting Lodging Resistance in Rice

**DOI:** 10.3390/genes12050718

**Published:** 2021-05-11

**Authors:** Bingxin Meng, Tao Wang, Yi Luo, Deze Xu, Lanzhi Li, Ying Diao, Zhiyong Gao, Zhongli Hu, Xingfei Zheng

**Affiliations:** 1State Key Laboratory of Hybrid Rice, Hubei Lotus Engineering Center, College of life sciences, Wuhan University, Wuhan 430072, China; 2016202040076@whu.edu.cn (B.M.); 2019202040069@whu.edu.cn (T.W.); 2019202040076@whu.edu.cn (Y.L.); ydiao@whu.edu.cn (Y.D.); zygao@whu.edu.cn (Z.G.); 2Institute of Food Crops, Hubei Academy of Agricultural Sciences, Wuhan 430064, China; dezexu@163.com; 3Hunan Engineering Technology Research Center, Hunan Agricultural University, Changsha 410128, China; lancy0829@163.com

**Keywords:** lodging resistance, rice, stem, leaf type, GWAS, gene-based association analysis

## Abstract

Lodging reduces rice yield, but increasing lodging resistance (LR) usually limits yield potential. Stem strength and leaf type are major traits related to LR and yield, respectively. Hence, understanding the genetic basis of stem strength and leaf type is of help to reduce lodging and increase yield in LR breeding. Here, we carried out an association analysis to identify quantitative trait locus (QTLs) affecting stem strength-related traits (internode length/IL, stem wall thickness/SWT, stem outer diameter/SOD, and stem inner diameter/SID) and leaf type-associated traits (Flag leaf length/FLL, Flag leaf angle/FLA, Flag leaf width/FLW, leaf-rolling/LFR and SPAD/Soil, and plant analyzer development) using a diverse panel of 550 accessions and evaluated over two years. Genome-wide association study (GWAS) using 4,076,837 high-quality single-nucleotide polymorphisms (SNPs) identified 89 QTLs for the nine traits. Next, through “gene-based association analysis, haplotype analysis, and functional annotation”, the scope was narrowed down step by step. Finally, we identified 21 candidate genes in 9 important QTLs that included four reported genes (TUT1, OsCCC1, *CFL1*, and *ACL-D*), and seventeen novel candidate genes. Introgression of alleles, which are beneficial for both stem strength and leaf type, or pyramiding stem strength alleles and leaf type alleles, can be employed for LR breeding. All in all, the experimental data and the identified candidate genes in this study provide a useful reference for the genetic improvement of rice LR.

## 1. Introduction

Rice (*Oryza sativa* L.) is one of the three main crops, meaning that the yield of rice is crucial to food security. However, lodging is a major problem in rice production, which not only leads to serious yield reduction, but also reduces the quality of rice [1,2]. In the 1960s, plant breeders reduced the risk of lodging by introducing a semi-dwarf gene *sd1*, known as the green revolution gene [3,4]. However, dwarfism also limits canopy photosynthesis, biomass, and food production [2,5]. At present, high-yielding varieties with large spikes make the stalks prone to lodging, especially when it is raining or windy [6]. In field management, high planting density and high fertilizer input are widely used, which also leads to lower stalk strength and higher lodging risk [7]. 

Both stem strength and leaf type-related traits are complex quantitative traits, which are affected by many factors, such as culm morphology, stem diameter, internode length and stem wall thickness, flag leaf morphology, length and angle, chlorophyll content, and environmental conditions. Studies have shown that the physical strength of stem is highly correlated with lodging resistance of rice [8,9,10,11,12]. Similarly, many previous studies have shown that yield is mainly affected by leaf type [13]. Some breeding trials have attempted to exploit strong-stem alleles, but these efforts have not been very successful because of the negative trade-offs between stem strength and grain yield [6]. One promising solution to this problem would be to identify alleles that increase stem strength without having a negative effect or even have a positive effect on yield.

To date, many genes governing stem strength have been identified and cloned, such as *SCM2*, which is identical to *APO1* and was previously reported to control culm diameter [14]. *BC1* encodes a COBRA (COBRA, a glycosyl-phosphati-dyl inositol-anchored protein/GPI anchored protein, control the correct localization of cell wall cellulose microfilaments and cell directional elongation) protein gene that regulates thickness of cell walls and physical strength of plants [15]. *SCM3* controls culm morphology [10]. *TUT1* allele *es1-1* showed shorter plant height and shorter internodes [16]. *OsCCC1* is a cation-chloride cotransporter gene, which is related to internode length [17,18]. In addition to these genes, many genes that affect leaf type have also been identified, such as *OsZHD1* [19], *CFL1* [20], *NAL1* [21,22,23,24], *RLI1* [25], *YGL1* [26], *OsAGO7* [27], *ACL1* [28], *BGL11(t)* [29], *RLS3* [30], and *REL2* [31].

Stem strength and leaf type are closely related to lodging resistance and yield. Up to now, there have been few reports on the genetic basis of stem strength and leaf type of rice using many rice germplasm resources. In the present study, we carefully selected 550 varieties and measured phenotypic data related to stem and leaf type over two years. Moreover, more than four million high-quality SNPs were obtained by re-sequencing. Then, following genome-wide association study, a combination of screening methods was used to identify candidate genes associated with rice stem strength and leaf type. Therefore, our experimental data and the identification of new candidate genes not only enrich the genetic resources of lodging resistance breeding, but also provide a new perspective.

## 2. Materials and Methods

### 2.1. Materials

We have collected more than 2000 rice germplasms from all over the world, to minimize the influence of heading dates on rice stem strength and leaf type traits to be measured and ensure the genetic diversity of the population. Finally, 550 rice germplasms were selected to use as the materials in this study. Among them, 327 accessions were selected from the 3K Rice Genome Project (3K RGP) [32] (Appendix A). 

### 2.2. Field Trials and Trait Measurements

We conducted the field experiment for two years, from mid-May to late-September in 2019 and 2020. The experimental field is in the experimental fields of the Institute of Food Crops, Academy of Agriculture Sciences, Hubei Province, China. About 25 d after germination, plants were transplanted into the field. Each accession was planted in a five-row, six-column plot at a spacing of 20 × 25 cm^2^ for each accession. In terms of field management, we were in line with local standard management practices. At the heading stage (more than 50% of the individuals in the population have headed), four plants were harvested randomly in the middle of each planting interval to measure leaf type-related traits (flag leaf length/FLL, flag leaf angle/FLA, flag leaf width/FLW, leaf-rolling/LFR, and SPAD). At two weeks after heading, four traits associated with stem strength were measured, including internode length (IL), stem wall thickness (SWT), stem outer diameter (SOD), and stem inner diameter (SID). FLL, FLW, and IL were measured with a ruler; We measured the FLA with an electronic protractor (FLA is the angle between the flag leaf and the stem); SPAD-502 was used to measure the base, middle, and front parts of flag leaves, and the average value was used for estimated SPAD value. Using vernier calipers to measure SWT, SOD, and SID, it is measured in the middle of the second internode (counted from the base). After measuring the leaf width (W) at the widest point of the flag leaf, then measuring the distance (w) between the leaf margins of the curled leaf here, LFR = (W − w)/W. The average trait values of each accession were used in data analyses of GWAS.

### 2.3. Genotyping

First, using the Illumina Hiseq X10 platform we obtained numbers of paired-end 150-bp reads. Then, the original sequence is further processed to eliminate adapter contamination and poor-quality reads. The process resulted in 4 GB data for each accession. Software BWA was used to map against the reference genome: Os-Nipponbare-Reference-IRGSP-1.0 [33]. SNP calling was conducted using GATK [34]. If there are more than two alleles in an SNP site, only the first two most important alleles are retained, and all subsequent minor alleles are considered missing. Not only that, but if a SNP site was heterozygous, it was also to be considered missing. If the missing rate of a SNP loci is greater than 20% or the minor allele frequency (MAF) is less than 5%, it will be removed. IMPUTE2 is used for imputing missing genotypes [35]. Finally, a total of 4,076,837 SNPs were used in the GWAS.

### 2.4. Population Structure and Kinship Analysis

All the SNPs were sampled to calculate population structure (Q) and kinship (K). We choose Principal Component Analysis/PCA method to calculate population structure. The Principal Components and kinship (K) of the varieties were performed using a R Package “rMVP” [36]. The Q and K matrix were used in the following association analysis.

### 2.5. Linkage Disequilibrium Analysis

The software “PopLDdecay” was used to calculate the linkage disequilibrium (LD) between pairs of markers in the population by the *r*^2^ command, which squared the Pearson’s correlation coefficient (*r*^2^) [37]. The marker pairs were divided into 5 kb uniform windows, and the average *r*^2^-value is used to estimate the *r*^2^-value of the window. The LD decay rate was evaluated by chromosome distance. When the mean value of r^2^ decreased to half of the maximum value, the distance traveled by chromosome was just the LD decay distance [38].

### 2.6. Genome-Wide Association Study and Candidate Genes Identification

We combined genotype data (single-nucleotide polymorphisms/SNPs), phenotypic data (Mean value of character), and covariates (Q and K matrix) to perform GWAS to detect the significant trait associated SNPs of all detected traits in both environments, loading the R package “rMVP” and input the genotype and phenotype data above. The general linear model (GLM), the mixed linear model (MLM) [39], and the Fixed and Random Model Unification (FarmCPU) [40] were performed. MLM used the Q and K matrix to adjust for cryptic relationships and other fixed effects. The Bonferroni multiple testing correction was applied to identify significant markers. Quantile–Quantile plots (Q-Q plots) and Manhattan plots with threshold lines were performed also using the R package “rMVP”. When the test statistics of a SNP reached *p* < 2.45 × 10^−7^ (*p* = 1/n, n: the number of SNP) in at least one of the two environments, we claimed that the SNP affected the measured traits.

Next, considering LD decay distance, we defined the interval of significantly associated SNP ± LD decay distance (125 kb) as a QTL. To reduce QTL redundancy, if there is overlap between QTL areas, they are combined into one QTL [41,42,43]. Then, as long as one of the following conditions can be met, we call it an important QTL: first, repeated detection within two years; second, close to the cloned gene; third, it contains successive distinct peaks; fourth, it affects multiple traits [44]. Once we have identified the candidate regions, the next step is to screen candidate genes by calculating the polymorphism effect of the nucleotides. This requires us to group the gene polymorphisms within the candidate intervals, and in this experiment we focus on two types of groupings. Group I: SNPs with significant *P* values and resulting in amino acid or splicing mutations. Group II: SNPs with significant *P* values and located near the 5′ end of the gene (≤2 kb from the first ATG, for example, promoter region) [45]. Then, we used these polymorphic sites for haplotype analysis, and genes with significant differences between haplotypes were identified as candidate genes. Finally, we also need to make functional annotations for these candidate genes to further determine whether they are related to the corresponding investigated traits.

## 3. Results

### 3.1. Phenotypic Variation and Correlations

In general, the phenotypes of most of the traits were normally distributed, but some of the traits were skewed distributions, especially for leaf-rolling. (Figure 1A). All the investigated traits showed high phenotypic diversity, suggesting that this population contained a rich and diverse genetic basis. In contrast to other traits, the phenotypes distributed of flag leaf length showed significant differences among the two years, indicating that the environment had a great influence on it (Figure 1A). The correlation between the measured traits also warrants attention and analysis. SPAD was negatively correlated with flag leaf length, internode length, and stem diameter. Flag leaf angle was positively correlated with internode length. Although positive correlations were observed between flag leaf length, flag leaf width, and all the stem strength-related traits, suggested there was a synergistic effect between stem strength and leaf type (Figure 1B). Overall, there was a strong correlation between stem strength and leaf type-related traits. 

### 3.2. Statistics of Markers

We have a total of 4,076,837 high-quality SNPs distributed across 12 chromosomes. The number of markers on each chromosome varies, with chromosome 9 having the least (260,134) markers and chromosome 1 having the most (457,835). The size of chromosome varied from 22.9 Mb for chromosome 9 to 43.2 Mb for chromosome 1. In addition, the mean distribution space of SNPs on chromosomes ranged from 0.076 kb on chromosome 11 to 0.106 kb on chromosome 5 (Table 1). Overall, the SNPs are evenly distributed, the number of SNPs within 1 Mb window size almost near 10,000 (Figure 2). 

### 3.3. Population Structure, Kinship and LD Patterns

All high-quality SNPs were used for principal component analysis to quantitatively analyze the population structure. As can be seen from the principal component score plot, the score points are continuously distributed without obvious clustering, indicating that the population we selected does not have complex population structure relations (Figure 3A). There was only one major subpopulation in the current panel according to the results of the PCA plot and kinship (Figure 3A,B). LD attenuation is another important factor in determining the efficiency of GWAS. When the LD maximum value decays to half, the corresponding physical distance is around 40 Kb (Figure 3C). Compared with previous reports, we have a smaller LD attenuation distance, which is favored for us to locate the candidate genes.

### 3.4. Identification of Significant Loci for Related Traits through GWAS

For all the traits investigated, we identified a total of 89 QTLs within two years, ranging from three QTLs for flag leaf width to as many as 33 QTLs for LFR. Among them, 52 (61) QTLs were detected only in 2019, 28 (37) QTLs were detected only in 2020, and 9 QTLs were commonly identified in two years (Table 2). 

Thirty-three QTLs for leaf-rolling were detected on all chromosomes except chromosome 1. Twenty-three QTLs were detected only in 2019 including *qLFR2.1*, *qLFR2.2*, *qLFR2.3*, *qLFR3.2*, *qLFR3.3*, *qLFR3.4*, *qLFR3.5*, *qLFR4.2*, *qLFR4.3*, *qLFR4.5*, *qLFR5.1* etc. Eight QTLs were detected only in 2020 including *qLFR2.1*, *qLFR3.1*, *qLFR4.1*, *qLFR4.4*, *qLFR5.3*, *qLFR6.1*, *qLFR6.3*, *qLFR7.3*, *qLFR8.3*, *qLFR10*. Two QTLs were identified in both 2019 and 2020 including *qLFR2.1*, *qLFR5.3* (Table 2). 

For flag leaf width, 3 QTLs were detected on chromosomes 2, 5, and 8. Two QTLs *qFLW2*, *qFLW3* were detected only in 2020, One QTL *qFLW8* was detected in both years. Four QTLs affecting leaf-rolling were detected on chromosomes 2 and 5 only in 2020 including *qFLA2.1*, *qFLA2.2*, *qFLA5.1*, *qFLA5.2*. For SPAD, Six QTLs were identified on chromosomes 5, 6, 7, 10, 11. Two QTLs were detected only in 2019 including *qSPAD7*, and *qSPAD10*. Four QTLs were detected only in 2020 including *qSPAD5*, *qSPAD6*, *qSPAD11.1*, and *qSPAD11.2* (Appendix A).

For stem wall thickness (SWT) 16 QTLs were detected on all chromosomes except 2, 10 and 11. Eleven QTLs, *qSWT1*, *qSWT3*, *qSWT4.1*, *qSWT4.2*, *qSWT4.4*, *qSWT6*, *qSWT7.1*, *qSWT7.2*, *qSWT8*, *qSWT9*, and *qSWT12*, were detected only in 2019. Two QTLs, *qSWT5.1* and *qSWT5.3* were detected only in 2020. Three QTLs were identified in both years including *qSWT4.3*, *qSWT5.2* and *qSWT7.3*. For internode length (IL), ten QTLs were identified on chromosomes 1, 4, 6, 8, 9. Five QTLs were detected only in 2019 including *qIL1.1*, *qIL6*, *qIL8.3*, *qIL8.4*, and *qIL9*. Three QTLs were detected only in 2020 including *qIL4.1*, *qIL4.2* and *qIL8.2*. Two QTLs *qIL1.2* and *qIL8.1* were detected in both years (Table 2).

A total of 17 stem inner diameter QTLs were detected on all chromosomes except chromosome 4 and 11. Eleven QTLs were detected only in 2019 including *qSID2*, *qSID3*, *qSID5.1*, *qSID5.2*, *qSID6.2*, *qSID8.2*, *qSID9.1*, *qSID9.2*, *qSID10.1*, *qSID10.2* and *qSID12.1*. Five QTLs were detected only in 2020 including *qSID1*, *qSID6.1*, *qSID6.3*, *qSID7* and *qSID8.3* (Appendix A)

### 3.5. Screening and Haplotype Analysis of Candidate Genes in Important QTL Regions

We carefully selected 9 important QTLs of 5 traits including leaf-rolling, SPAD, internode length, stem wall thickness, and stem inner diameter for further analysis. They were four QTLs close to the reported gene including *qLFR2.3*, *qLFR9.1*, *qIL1.1* and *qIL8.2*. Three QTLs that were repeatedly identified within two years including *qIL8.1*, *qSWT5.2* and *qSID8.1* (Table 2), and two QTLs *qSPAD7* and *qSPAD11.1* containing successive significant Manhattan peaks. In addition, *qIL8.1* and *qSID8.1* have a large overlap region, and both belong to a same pleiotropic QTL. We performed haplotype analysis using the two groups of polymorphisms (see method above). Finally, we found 21 candidate genes, ranging from one to nine candidate genes for each region (Appendix A).

For internode length (IL), three important QTLs *qIL1.1*, *qIL8.1* and *qIL8.2* were selected for candidate gene analysis. There were 47, 21, and 8 genes in *qIL1.1*, *qIL8.1* and *qIL8.2* regions, respectively. In addition, there were 38, 19, and 5 genes have significant mutations in *qIL1.1*, *qIL8.1* and *qIL8.2*, respectively. 233 polymorphisms in *qIL1.1*, including 226 nonsynonymous, 5 stop-gain and 2 stop-loss. In *qIL1.1*, we found only *Os01g0208600* have significant phenotypic differences between the haplotypes, which is identical to *TUT1* [16,46]. Five major haplotypes of *TUT1* were found (*n* ≥ 6, n: the number of individuals of each haplotype). Haplotype AGAATG corresponded to the reference genome and was associated with significantly longer internode length than other haplotypes (Figure 4a), which is consistent with the previous study on *TUT1*. Inside *qIL8.1*, there are 76 nonsynonymous and 1 stop-gain. Significant differences were found between haplotypes of two genes *Os08g0243100* and *Os08g0244500* (Figure 4b). In *qIL8.2* regions, there were no continuous peaks significantly associated with the traits; however, there was a reported gene *OsCCC1* [18] about 29 Kb away from the Peak-SNP 14215369 (*p* = 4.22 × 10^−8^) (Table 2). Significant differences were found between haplotypes of *Os08g0323700*. which is identical to *OsCCC1*. Haplotype GCCCTCCCAGCAGTC has the longest internode length, while haplotype GCCCTCCCTGCAGTC has the shortest internode length (Appendix A).

For stem wall thickness (SWT), only one QTL *qSWT5.2* were selected for candidate gene analysis. There were 22 genes in *qSWT5.2* regions, 17 genes have significant mutations, two of which were premature termination of translation. Finally, nine genes showed significant differences between haplotypes. They are *Os05g0200160*, *Os05g0200340*, *Os05g0200400*, *Os05g0200500*, *Os05g0201300*, *Os05g0202200*, *Os05g0202300*, *Os05g0202550* and *Os05g0202600.* They have 4, 4, 4, 3, 3, 3, 4, 3 and 3 haplotypes, respectively (Figure 4c, Appendix A). Of the nine genes, *Os05g0200340* was less likely the candidate gene. Since a mutation within one haplotype (CCTG) that causes a premature termination of translation showed the same SWT phenotype as haplotype GTGG and GCGG which containing nonsynonymous mutation.

For stem inner diameter (SID), we choose QTL *qSID8.1* for analysis. However, there is a large overlap region between *qSID8.1* and *qIL8.1* (Table 2), suggesting that the overlap region may affect both SID and internode length simultaneously. In *qSID8.1*, there are a total of 20 genes, 76 nonsynonymous mutations and 1 stop-gain belonging to 9 genes, respectively. However, only two genes *Os08g0243100* and *Os08g0243500* ended up with significant differences in haplotypes. Eight haplotypes were found for *Os08g0243500*, and two haplotypes were found for *Os08g0243100*. We also found that *Os08g0243100* has a pleiotropic effect, affecting both stem inner diameter and internode length (Figure 4a,d). The results of the analysis have supported our previous hypothesis.

For leaf-rolling (LFR), two important QTLs *(qLFR2.3*, *qLFR9.1*) close to the reported genes were selected for analysis. Beyond our expectations, we found some interesting things. In *qLFR2.3*, performing haplotype analysis of polymorphisms in group II, we only found that haplotype of *Os02g0516400* was significantly different in 2020, and the changes of haplotype phenotype values were inconsistent between 2019 and 2020 (Appendix A). *Os02g0516400* is identical to *CFL1*, overexpression or inhibition of this gene can significantly affect leaf-rolling [20]. Interestingly, when we did haplotype analysis using polymorphisms in group I, we found that haplotype phenotypic values were consistent between two years. Haplotype GGCACCGCTACGT has the maximum LFR value compared to other haplotypes (Figure 4e). In *qLFR9.1*, there were 35 genes in total, among which 14 genes had amino acid mutation or premature termination of translation. Then, haplotype analysis revealed significant differences between three genes *Os09g0471000*, *Os09g0471100* and *Os09g0471200* (Figure 4f). *Os09g0466400* is identical to *ACL-D*, and the T-DNA insertion of *ACL-D* resulted in increased expression levels and leaf curl [19]. We found that its haplotype phenotypic value changed in the same trend in 2 years. Haplotype GAAAGTATRTG, always has the lowest phenotypic value. However, we did not find significant differences between its haplotypes (Appendix A). 

For SPAD, Two QTLs *qSPAD7* and *qSPAD11.1* were selected for analysis. In *qSPAD7*, there were a total of 34 genes, among which 19 genes had amino acid changes or premature termination of translation, while only *Os07g0192000* had a significant difference between haplotypes. *Os07g0192000* has only two haplotypes (Figure 4g). At coding site 1257, G is replaced by T, resulting in the change of glutamic acid at 419 to aspartic acid. The function of *Os07g0192000* is predicted to be ATPase. In the previous study, a cloned gene, *LMR*, encodes a type AAA ATPase, and affects the contents of chlorophyll A, chlorophyll B and carotenoids [47,48]. Therefore, we speculate that *Os07g0192000* is the most likely candidate gene for *qSPAD7*. In *qSPAD11.1*, there were a total of 30 genes, among which 17 genes had amino acid changes or premature termination of translation, while only *Os11g0682600* had a significant difference between haplotypes. It has five haplotypes, and haplotype CGACCT has a relatively small SPAD value (Figure 4h).

## 4. Discussion

Lodging resistance is an important trait for the high yield of rice [49]. A great deal of research has been done on lodging resistance breeding of rice. Nevertheless, rice breeding is always faced with the trade-offs between high yield and lodging resistance. Large spike varieties with high yield are prone to lodging [6]. Semi-dwarfing limits their yield potential [2,5], and dense planting with high fertilizer is prone to lodging when it is windy and rainy [7]. In previous LR breeding, most of the focus has been on the genetic basis of LR itself, ignoring the correlation between traits. However, in crop breeding, we should fully consider the relationship between different traits so that the target trait can be improved, and unnecessary losses can be reduced [50]. Recently, some researchers have found that stem strength is positively correlated with panicle weight, whereas stem strength is negatively correlated with panicle number. Therefore, they looked for genetic loci with pleiotropic effects on stem strength and yield, and eventually found and validated one gene: *OsPRR37* [51]. In our study, based on phenotypic data, we found a strong correlation between stem strength and leaf type-related traits. We proposed an introgression of alleles that are beneficial for both stem strength and leaf type, or pyramiding stem strength alleles and leaf type alleles for LR breeding. However, due to epistasis, the results of pyramiding these genes are difficult to predict. Therefore, in addition to discovering new genes, there is a need to investigate which alleles are the most favorable and how each gene is related to the others.

In our study, we carefully selected 550 varieties and measured 2-year phenotypic data related to stem strength and leaf type. More than 4 million high-quality SNPS were obtained through re-sequencing. Then, association analysis and haplotype analysis were carried out to identify candidate genes. We have shortlisted 21 candidate genes governing 9 important QTLs affecting the investigated traits. These candidates included four cloned QTL genes governing internode length and LFR. The first one was *qIL1.1*, for which only one single candidate *gene Os01g0208600* was detected, which is identical to *TUT1*/*ES1**. TUT1* encodes an inhibitor of the cAMP-like receptor protein, and *es1-1* shows shorter plant height, shorter internode, and reduced seed setting. ES1-1 is also important for actin assembly and panicle development, and regulates water loss [16,46]. The second one was in *qIL8.2*, the candidate *gene Os08g0323700*, which is identical to *OsCCC1* [18]. *OsCCC1* is a cation-chloride cotransporter gene. Compared with the wild type, the mutant had a shorter stem, narrower leaf, thinner basal stem, and lower field yield. The third one was in *qLFR2.3*, a rice dominant leaf-rolling mutant, *CFL1**(**Os02g0516400**)*, encoding the WW domain (WW domain is a coherent and compact domain generally composed of 38~40 amino acid residues. It features two tryptophan residues, and interacts specifically with proteins containing XPPXY conserved sequence.). The difference expression of CFL1 in Arabidopsis thaliana and rice can both affect the development of cuticle [20]. However, we found that mutations in the promoter region were not consistent with phenotypic changes (Appendix A). In contrast, haplotype analysis in group I found that mutations were consistent with phenotypic changes, and that there were significant differences between haplotypes (Figure 4f). The last one was in *qLFR9.1*, for which four candidate genes were identified. One of these genes was *ACL-D* (*Os09g0466400*) encoding a Zn-finger transcription factor. Compared to the control group, the leaves of *acl-d* mutants were curled, drooped to the distal end, the number of vesicular cells in the leaves of *acl-d* mutants increased, and the arrangement was abnormal [19].

The identification of the above four cloned genes indicated that our analytical method is reliable. Using the same analytical strategy, we also identified several new candidate genes, e.g., two candidate genes *Os08g0243100* and *Os08g0244500* for *qIL8.1*, of which the most likely one was *Os08g0243100* (dehydrogenase-phosphopantetheinyl transferase, putative, expressed). Based on the prediction of protein function, it is likely to affect the biosynthesis of fatty acids. However, a reported gene, *SSI2*, that reduced elongation specifically in the second internode counted from the top, encodes a fatty-acid dehydrogenase and affects the process of fatty-acid biosynthesis [52]. Based on the functional similarity between *Os08g0243100* and *SSI2*, we hypothesized that *Os08g0243100* is the causal gene of *qIL8.1*. Nine candidate genes *Os05g0200160* (Hypothetical gene), *Os05g0200340* (Hypothetical gene), *Os05g0200400* (cytochrome P450, putative, expressed), *Os05g0200500* (CK1_CaseinKinase_1a.5-CK1 includes the casein kinase 1 kinases, expressed), *Os05g0201300* (Hypothetical gene), *Os05g0202200* (expressed protein), *Os05g0202300* (lachrymatory factor synthase, putative, expressed), *Os05g0202550* (Hypothetical conserved gene), and *Os05g0202600* (expressed protein) for *qSWT5.2*. Because we have not found any stem wall thickness-related cloned genes with similar functions to these genes, this is the first time that the genes with these functions have been reported to be related to stem wall thickness. Two candidate genes were *Os07g0192000* (ATPase, putative, expressed) and *Os11g0682600* (rust-resistance protein Lr21, putative, expressed) for *qSPAD7* and *qSPAD11.1*, respectively. In the previous study, a cloned gene, *LMR*, encodes a type AAA ATPase, and affects the contents of chlorophyll A, chlorophyll B and carotenoids [47,48]. *Os07g0192000* and *LMR* encode proteins with the same function. Therefore, we speculate that *Os07g0192000* is the most likely candidate gene for *qSPAD7*. We will conduct transgenic experiments in the following studies to verify the function of the above candidate genes.

## 5. Conclusions

In this study, we carefully selected a natural population consist of 550 varieties. Through Genome-wide association study (GWAS), gene-based association analysis, haplotype analysis, and functional annotation, Finally, we identified 21 candidate genes. These candidate genes of new loci affect stem strength and leaf type-related traits, and our data provided a solid foundation for further research on the genetic improvement of rice lodging resistance and yield. 

## Figures and Tables

**Figure 1 genes-12-00718-f001:**
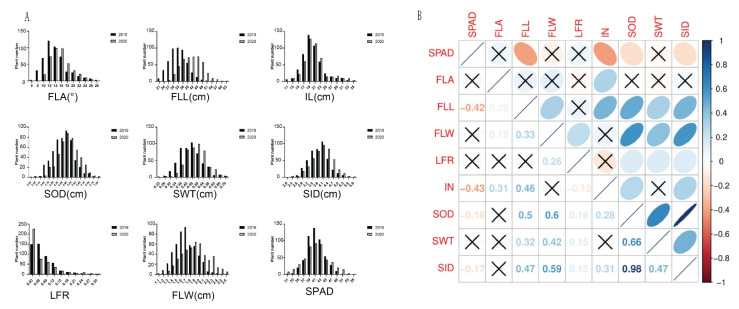
Phenotypic Variation and Correlations. (**A**) The bar chart of nine traits in two years. Black and gray color indicated 2019 and 2020, respectively; FLL, flag leaf length; FLW, flag leaf width; IL, internode length; LFR, leaf-rolling; SWT, stem wall thickness; SPAD; SID, stem inner diameter; SOD, stem out diameter; FLA, flag leaf angle. (**B**) Correlations between the mean values of the nine traits. The areas and colors of ellipses showed the absolute value of corresponding correlation coefficients (*r*) (upper triangular). Right and left oblique ellipses indicated positive and negative correlations, respectively. The values were corresponding *r* between the nine traits (lower triangular). The **×** indicated insignificant at 0.05.

**Figure 2 genes-12-00718-f002:**
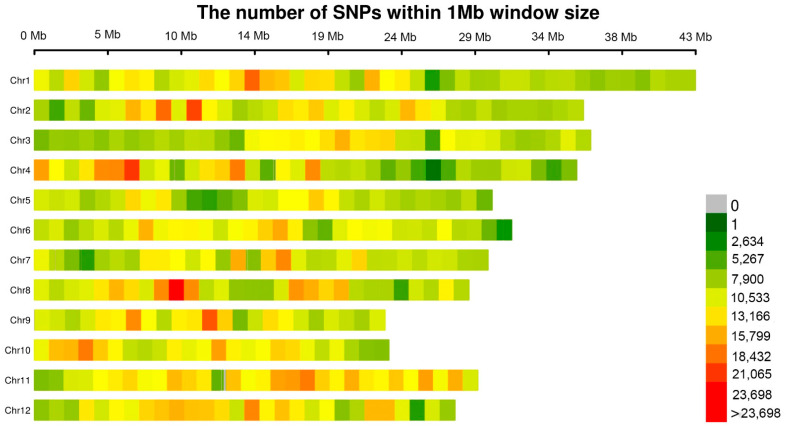
The number of SNPs within 1 Mb window size.

**Figure 3 genes-12-00718-f003:**
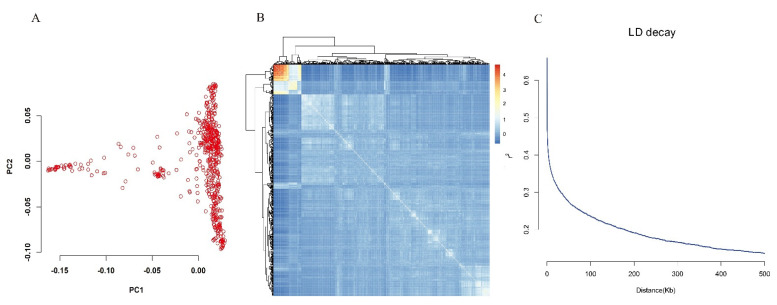
Population Structure, Kinship, and LD Patterns. (**A**) PCA plot for the 550 varieties based on whole-genome sequence data. PC1 and PC2 indicate score of principal components 1 and 2, respectively. (**B**) Heat map of kinship from R Package “pheatmap” with the tree shown on the top and left. (**C**) LD decay. Y-axis was the average *r*^2^ value of each 5 kb region and X-axis was physical distance between markers.

**Figure 4 genes-12-00718-f004:**
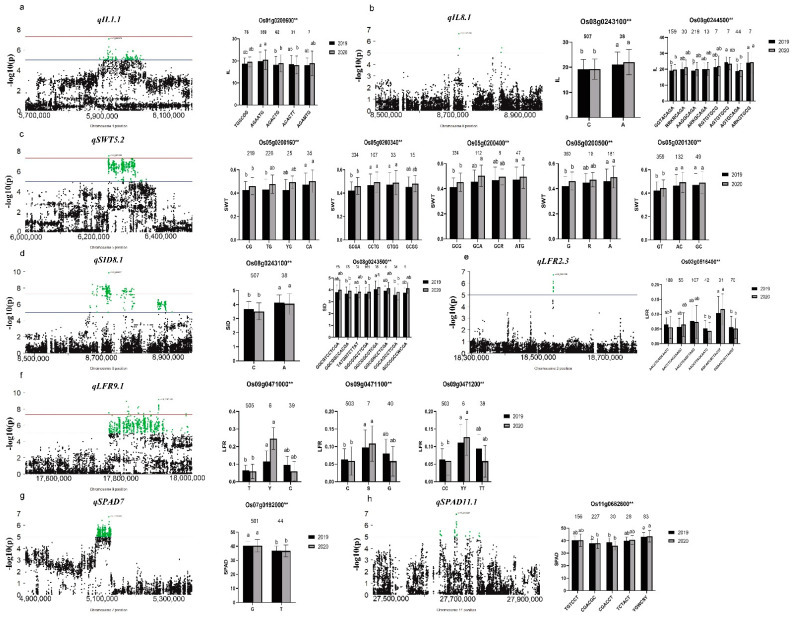
Gene-based association analysis of eight important QTL loci and haplotypes analysis of targeted genes of related QTL. Including *qIL1.1* (**a**), *qIL8.1* (**b**) *qSWT5.2* (**c**), *qSID8.1* (**d**), *qLFR2.3* (**e**), *qLFR9.1* (**f**), *qSPAD7* (**g**) and *qSPAD11.1* (**H**). Dash line showed the threshold to determine significant SNP. The ** suggested significance of ANOVA (for ≥three haplotypes) or *t*-test (for two haplotypes) at *p* < 0.01. The letter on histogram (a and b) indicated multiple comparisons result at the significant level 0.01. The value on the histogram was the number of individuals of each haplotype. Black and gray colors indicated 2019 and 2020, respectively.

**Table 1 genes-12-00718-t001:** Distributions of markers on chromosomes.

Chr	Marker No.	Size (Mb)	Spacing (kb)
chr1	457,835	43.2	0.094
chr2	387,459	35.9	0.092
chr3	361,044	36.3	0.101
chr4	368,017	35.5	0.096
chr5	280,079	29.7	0.106
chr6	333,313	31.1	0.093
chr7	306,057	29.7	0.097
chr8	335,328	28.4	0.085
chr9	260,134	22.9	0.088
chr10	282,524	23.1	0.082
chr11	380,739	29.0	0.076
chr12	324,308	29.4	0.091
Total	4,076,837	372.2	0.091

**Table 2 genes-12-00718-t002:** Some important QTLs identified for traits in 2019 and 2020 in Wuhan, China.

QTL	Year	CHROM	POS	REF	ALT	Effect	SE	*p*
*qLFR2.1*	2019	2	4,243,076	G	A	0.06	0.011	1.17 × 10^−7^
	2020	2	4,461,960	G	T	0.05	0.010	4.24 × 10^−8^
*qLFR2.3*	2019	2	18,545,189	A	G	0.03	0.005	1.82 × 10^−7^
*qLFR5.3*	2020	5	20,540,653	A	G	0.06	0.011	1.42 × 10^−8^
	2019	5	20,540,653	A	G	0.05	0.009	4.64 × 10^−8^
*qLFR9.1*	2019	9	17,764,668	C	A	0.04	0.006	3.06 × 10^−8^
*qSID8.1*	2020	8	8,673,481	G	A	0.28	0.050	4.97 × 10^−8^
	2019	8	8,722,341	C	T	0.35	0.053	1.32 × 10^−10^
*qIL1.1*	2019	1	5,920,879	C	T	1.85	0.341	8.34 × 10^−8^
*qIL1.2*	2020	1	32,933,806	G	T	1.65	0.299	6.10 × 10^−8^
	2019	1	32,934,166	T	C	1.61	0.300	1.39 × 10^−7^
*qIL8.1*	2019	8	8,396,436	C	T	1.50	0.283	2.32 × 10^−7^
	2020	8	8,717,396	C	T	1.75	0.334	1.84 × 10^−7^
*qIL8.2*	2020	8	14,215,369	G	A	1.70	0.304	4.22 × 10^−8^
*qSPAD7*	2019	7	5,119,605	G	A	−1.12	0.212	1.92 × 10^−7^
*qSPAD11.1*	2020	11	27,695,597	C	A	−1.54	0.284	1.00 × 10^−7^
*qSWT4.3*	2019	4	29,983,581	T	A	−0.03	0.008	1.59 × 10^−7^
	2020	4	30,459,506	A	G	−0.06	0.010	9.45 × 10^−8^
*qSWT5.2*	2020	5	6,243,695	C	T	0.02	0.004	1.27 × 10^−7^
	2019	5	6,245,355	C	T	0.01	0.003	1.97 × 10^−8^
*qSWT7.3*	2019	7	28,576,217	T	C	−0.07	0.013	2.06 × 10^−7^
	2020	7	28,576,224	T	C	−0.07	0.013	2.06 × 10^−7^
*qFLW8*	2020	8	22,563,133	T	C	0.22	0.041	8.67 × 10^−8^
	2019	8	22,849,319	C	T	0.21	0.038	4.24 × 10^−8^

## Data Availability

The raw data for this study can be found in the BioProject ID PRJNA321462, PRJNA331215 on NCBI and 3k RGP. The URL is https://www.ncbi.nlm.nih.gov/sra, accessed date: 13 May 2016 and https://registry.opendata.aws/3kricegenome/, accessed on 25 July 2016.

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
