# Peer review of "Genome-Wide Association Study Identified Novel Candidate Loci/Genes Affecting Lodging Resistance in Rice"

_genes, 2021, doi:10.3390/genes12050718_

Round 1

Reviewer 1 Report

In this manuscript, the author did a genome-wide association study that identified novel candidate loci/genes affecting lodging resistance in rice. Authors have carried out an association analysis to identify quantitative trait locus affecting stem strength-related traits and leaf type associated traits using a diverse panel of 550 accessions and evaluated in two years. A Genome-wide association study using 4076837 high-quality single nucleotide polymorphisms identified 89 QTLs for the nine traits. Next, through "gene-based association analysis, haplotype analysis, and a functional annotation," the scope was narrowed down step by step; at last, we identified 21 candidate genes in 9 significant QTLs, that including four reported genes (TUT1, OsCCC1, CFL1, and ACL-D), and seventeen novel candidate genes. Introgression of alleles beneficial for both stem strength and leaf type, or pyramiding stem strength alleles and leaf type alleles can be employed for LR breeding.  The manuscript has some grammatical errors and some plagiarism in this manuscript. However, the study has some solid data, and I found no major drawback in this study. I have some suggestions for the betterment of this manuscript.

Please make some hypothetical last figure explaining mechanism, and that can accumulate this study finding along with already previously reported studies.

Change at

L47 yield mainly affected to yield is mainly affected.

L79 two years, during mid-May to two years, from mid-May.

L97 only that, but to only that but.

L147 distributions especially for LFR to distributions, especially for LFR.

L153 While, positive correlations were observed to While positive correlations were observed.

L154 between FLL, FLW and all to between FLL, FLW, and all.

Write down full form of SPAD.

L183 What is PCA? Write down full form.

L191 What is pheatmap?

L299 had significant difference to had a significant difference.

L308 color indicated to colors indicated.

L310 for high yield of rice to for the high yield of rice.

L320-322 rearrange this line. It have 2 time and.

L337 had shorter stem to had a shorter stem.

L348 acl-d to italic acl-d.

L372 under way to verify the functionalities of above candidate genes to underway to verify the functionalities of the above candidate genes.

I have found plagiarism at many parts at L62, L94, L110-111, L121-123, L127-129, L135-139, L151-152, L169-170, L181-182, L189-193, L239-240, L302-303, L304-306, L316-318, L372. Please clean it.

Reviewer 2 Report

Congratulations for completing such a big and complicated study. It is good for knowledge base, however, I would recommend going through the manuscript and report only necessary information. Please also improve english grammar. 
You have list many abbreviations. It is confusing at times and harder to follow, I would recommend reducing them if possible. 
You did not define how you measured these different traits. 
You need to improve quality of your figures and tables need to be self-explanatory. 

Line73-77: Can you please elaborate the criteria of selecting 550 rice germplasm from 2,000 lines. I understand that you considered heading dates but it is not clear on sentence “To maintain…”. If you defined criterial of heading date, please mention it

Line 82: “transplant to field plant” to “transplanted into the field”
Line 82: what is “col”, if it is column, please use full spelling 
Line 83: “we are” to “we were”
Line 93: Think about rephrasing this sentence “Finally, yielding more than 4 Gb data for each accession” to “The process resulted in 4 GB data for each accession.” 
Line 97: “an SNP site is heterozygous, it will also be” to “a SNP site was heterozygous, it was also be”
    I am not going to correct every sentence for tense related mistakes going forward. Please use past tense for the activities that you already performed. You used past tense in some sentences but not in other. For example – “Finally, a total of 4,076,837 SNPs were 100 used in the GWAS.” This is past tense and was used correctly but other sentences were not in the same tense. 
Line 105: use of article “the” before r package is not correct, should be “a”. Similarly going forward, please correct the article use, I will not pin point going forward with this review
Line 118: “Load…”, not a complete sentence
Line 120: you mentioned “respectively”, but it is not correct in sentence that it is respective of what 
Line 150: “During two years” to “between two years” or “among two years”
Fig.1A: Numbers are too small, not clear to read, improve quality
What is the difference between LFR and SPAD
You used many abbreviations, which is fine, but it is confusing at times, you would recommend to reduce them if possible. I am not sure what is IN and SD trait mean
FLR, I don’t see the definition of it, same as SPAD
What is PVE (percent variation explained) by these QTLs you reported in Table2. What is effect and SE in table2. If one of them is explaining phenotype, most of these QTLs did not have any major effect. I am not sure why do you have to report all of these in the main manuscript. You can include major QTLs in the main manuscript and rest of them in additional files, you can avoid many of these low effect QTLs by changing your criteria to define a QTL. 
Line237: “What’s more” – does not fit here
Line 310-311: “There has been……” sentence not clear
Line323-325: It is easy to state that statement but harder to do. Most breeding program focus on lodging resistance along with yield traits, however it is harder to focus on allele pyramiding. A breeder rather look at his line in the field and select for lodging resistance. If a main focus of a breeding program is to develop a lodging resistance variety then a breeder can do what you are recommending but it is not common. 

Round 2

Reviewer 1 Report

I am happy with the author revision and Manuscript looks better and polished than before. Manuscript can be accepted in its current format.

Reviewer 2 Report

Thank you for addressing the most of the comments. I would still suggest to improve figure quality. 

Good Luck